# Left ventricular trabeculation in *Hominidae*: divergence of the human cardiac phenotype
Bryony A. Curry[1], Aimee L. Drane [2,3] ✉, Rebeca Atencia[4], Yedra Feltrer[2], Thalita Calvi[5], Ellie L. Milnes [6,7,8], Sophie Moittié[9,10], Annika Weigold[11], Tobias Knauf-Witzens[11], Arga Sawung Kusuma[12], Glyn Howatson[13,14], Christopher Palmer[15], Mike R. Stembridge[16], John E. Gorzynski [17], Neil D. Eves[1], Tony G. Dawkins [1] & Rob E. Shave [1] ✉

Although the gross morphology of the heart is conserved across mammals, subtle interspecific variations exist in the cardiac phenotype, which may reflect evolutionary divergence among closely-related species. Here, we compare the left ventricle (LV) across all extant members of the *Hominidae* taxon, using 2D echocardiography, to gain insight into the evolution of the human heart. We present compelling evidence that the human LV has diverged away from a more trabeculated phenotype present in all other great apes, towards a ventricular wall with proportionally greater compact myocardium, which was corroborated by post-mortem chimpanzee (*Pan troglodytes*) hearts. Speckle-tracking echocardiographic analyses identified a negative curvilinear relationship between the degree of trabeculation and LV systolic twist, revealing lower rotational mechanics in the trabeculated non-human great ape LV. This divergent evolution of the human heart may have facilitated the augmentation of cardiac output to support the metabolic and thermoregulatory demands of the human ecological niche.

Mammals are a remarkably diverse class of vertebrates, capable of inhabiting every major biome on the planet. This diversity is associated with a vast range of environmental stressors and interspecific differences in posture and locomotion, creating very different hemodynamic challenges. Despite this remarkable diversity, the gross structure of the mammalian heart is highly conserved across species; retaining four chambers and a complete interatrial and interventricular septum[1].

Although the gross structure of the mammalian heart is conserved, interspecific features exist. For example, heart shape varies considerably across species, from broad and flat in whales to long and narrow in terrestrial ungulates[2]. Variation in the cardiac phenotype is also present among closely-related mammals[2], indicative of evolutionary divergence. While comprehensive data examining cardiac structure and function across the entire *Hominidae* taxon do not exist, preliminary work suggests that the left ventricle (LV) of adult male chimpanzees (*Pan troglodytes*) may be morphologically distinct from that of humans[3]. Prominent myocardial trabeculations, characterized by protrusions of the endocardium into the LV cavity with intertrabecular recesses, were previously observed in adult male chimpanzees[3]. This trabeculated phenotype differs from the relatively smoother ventricular wall typically observed in healthy humans[4], suggesting

that there may have been species-specific selective pressures on the heart during the evolution of *Hominidae*[3].

Cardiac morphology and function are closely linked[5]; therefore, the discrete structural attributes of the chimpanzee and human LV likely coincide with differences in systolic and diastolic ventricular function. Such interspecific cardiac phenotypes may be the result of selection for the hemodynamic demands associated with each species' ecological niche (i.e., the habitat and the role a species plays within an ecosystem). Indeed, previous data has shown that resting metabolic rate[6], physical activity and daily locomotion[7] are far greater in humans in comparison with other great apes, and so it is not surprising that cardiac output is also comparatively higher in humans[3]. The larger cardiac output in humans is likely supported by comparatively greater LV systolic and diastolic function (e.g., myocardial rotation and deformation), including LV twist[3]. LV twist, which is dependent upon the helical angulation of the aggregated cardiomyocytes[8–10], is characterized by counter-directional rotation of the LV base and apex during systole. Together with the velocity of LV untwisting during diastole, LV twist helps facilitate efficient filling and ejection of the ventricle, especially during periods of heightened metabolic and thermoregulatory demand[11,12].

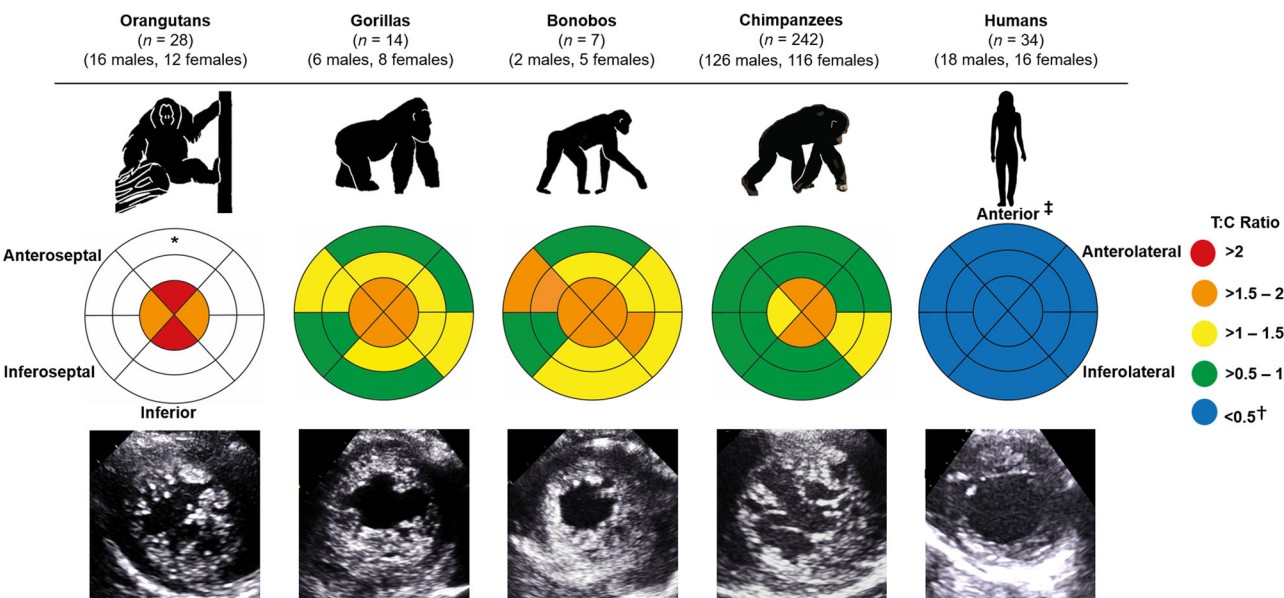

**Fig. 1 | Comparison of left ventricular trabeculation in great apes.** The bullseye plots represent the trabecular:compact (T:C) ratio for each segment of the left ventricle. The outer layer of the bullseye plots represents the basal segments, the middle and innermost layers represent the midpapillary and apical segments of the left ventricle, respectively. Red segments correspond to an average T:C ratio of >2; orange segments correspond to an average T:C ratio >1.5–2; yellow segments correspond to an average T:C ratio of >1–1.5; green segments correspond to an average T:C ratio of >0.5–1; blue segments correspond to an average T:C ratio of <0.05. Echocardiographic images of the parasternal short-axis at the apex are shown at end-diastole. *No data were available for the basal or midpapillary segments in the orangutans due to artifact from laryngeal air sacs. †These T:C ratios compare favorably with other reports in healthy human cohorts, ranging from 0.2 to 0.9[65,66]. ‡Anatomical labels have been provided in accordance with the conventional guidelines for cardiac chamber quantification by the American Society of Echocardiography and European Association of Cardiovascular Imaging[52]. However, we note that this clinical convention does not align with the recognized anatomical approach and may result in confusion across disciplines—see ref. 67 for further clarification.

The functional advantages associated with a LV capable of greater twist and untwisting velocity, combined with the preliminary data in adult male chimpanzees[3], prompt the hypothesis that the human heart has diverged from a trabeculated ancestral phenotype to support the specific metabolic and thermoregulatory demands of the human niche. To test this hypothesis, we compared LV structure across all extant great apes using 2D echocardiography and further explored trabeculation in a subset of post-mortem chimpanzee (*Pan troglodytes*) hearts. We then compared LV rotation and deformation between human and non-human great apes to explore whether the trabeculated phenotype is associated with differences in LV systolic and diastolic functional mechanics. Our findings point to evolutionary divergence of the human LV away from the phenotype of all other non-human great apes, which may have had important implications for cardiac function in early humans.

## Results

### Left ventricular trabeculation in great apes
Using 2D echocardiographic views of the LV, trabeculation was assessed in a mixed-sex and mixed-age sample of chimpanzees (*Pan troglodytes*; $n = 242$), orangutans (*Pongo;* $n = 28$), gorillas (*Gorilla gorilla;* $n = 14$), bonobos (*Pan paniscus;* $n = 7$) and humans ($n = 34$). The ratio of the trabecular to compact myocardium (T:C ratio) was calculated at end-systole for all LV segments and indicated that the LV of non-human great apes had a greater degree of trabeculation than that of humans ($P < 0.001$ to $P = 0.013$; Kruskal-Wallis test; Supplementary Table 1). This was particularly noticeable at the LV apex (Fig. 1), where the average T:C ratio of non-human great apes was approximately four times that observed in humans (Supplementary Table 1). To support our measurements from echocardiography, we examined post-mortem hearts from 15 chimpanzees that died of non-cardiac related causes (male $n = 11$, female $n = 4$; age range at death: 7–17 years; Supplementary Fig. 1). Post-mortem analyses confirmed the presence of ventricular trabeculations which, similar to our echocardiographic data, appeared to be greater in the apical region

(Supplementary Fig. 1). The extent of trabeculation was largely similar across non-human great apes, however, gorillas and orangutans had a greater degree of trabeculation compared with chimpanzees in a limited number of LV segments, although we note these data are from relatively small cohorts (Fig.1; Supplementary Table 1). Collectively, these results indicate that non-human great apes have a more trabeculated LV phenotype relative to that of humans, which typically have a proportionately greater compact to trabecular myocardium.

As previous work only examined adult male chimpanzees[3], we further investigated whether LV trabeculation varied across age and sex in our large cohort of chimpanzees to determine the ubiquity of this phenotype. Trabeculation did not differ between infant (≤4 years), juvenile (5–7 years), sub-adult (8–11 years), and adult (≥12 years) male and female chimpanzees (Fig. 2), at any level of the LV (all $P > 0.05$; two-way ANOVA; Supplementary Table 2). With the exception of the mid-inferior segment, which was greater in adult males compared to females ($P = 0.038$; two-way ANOVA), there were also no sex differences at either the basal or mid-papillary level of the LV for any of the age groups (all $P > 0.05$; two-way ANOVA; Fig. 2; Supplementary Table 2). However, the average T:C ratio across the four segments of the LV apex was greater in adult male chimpanzees compared with their female counterparts (on average, 6%, $P = 0.001$; two-way ANOVA; Supplementary Table 2).

### Left ventricular rotation and deformation
2D speckle-tracking echocardiography was used to investigate potential differences in LV rotation and deformation between human and non-human great apes. Since LV function varies with age[13], and due to the sample sizes of our great ape populations, this hypothesis was tested in adult chimpanzees ($n = 136$) and adult humans ($n = 34$). Consistent with preliminary data from adult male chimpanzees[3], systolic apical rotation was lower in both male and female chimpanzees compared with their human counterparts (on average, 69%; Supplementary Table 3; incorporated into Fig. 3), resulting in less LV twist during systole (on average, 50%; all

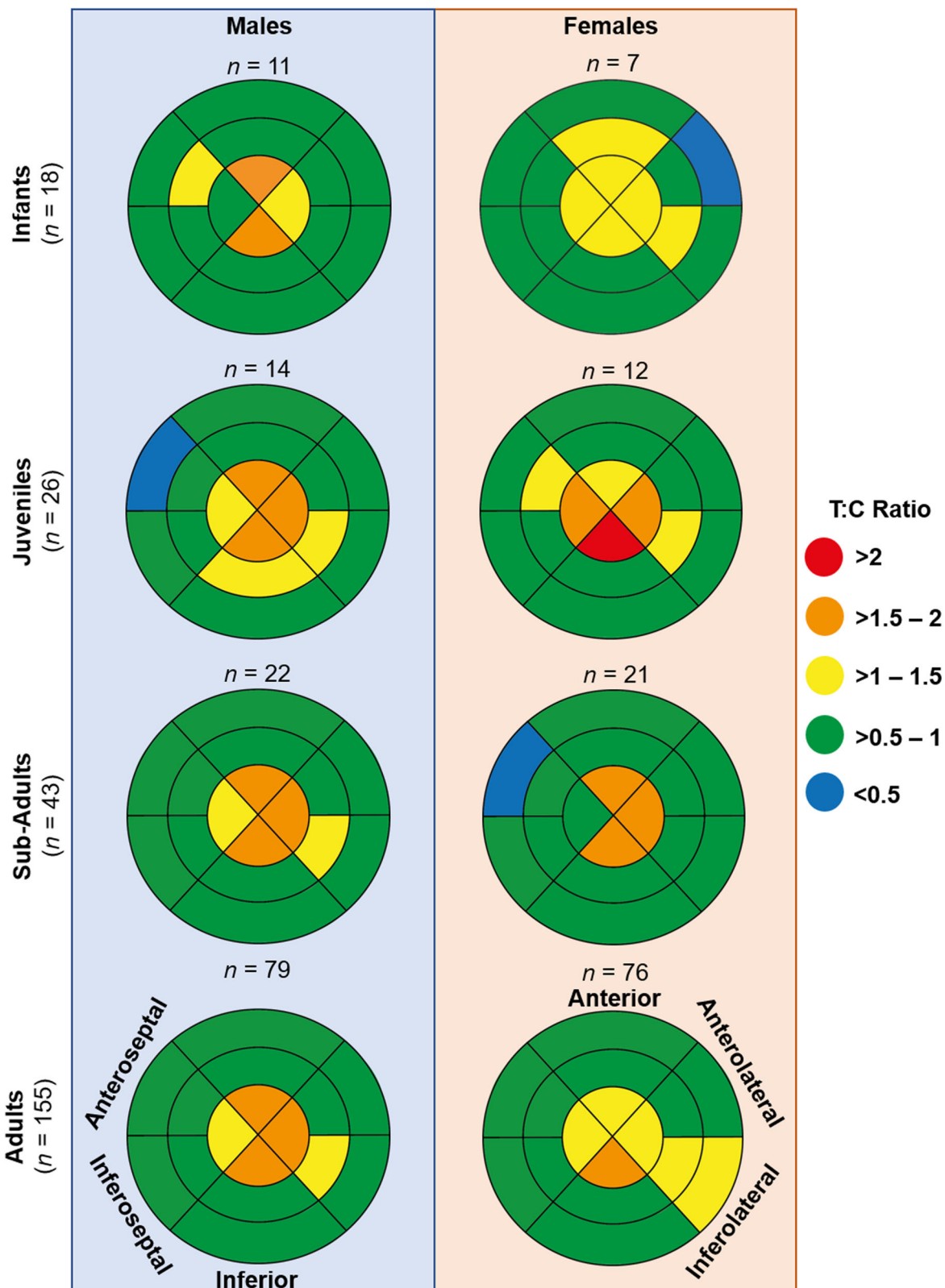

**Fig. 2 | Graphical representation of the trabecular:compact (T:C) ratio for each segment of the left ventricle in chimpanzees.** The outer layer of the bullseye plots represents the basal segments, the middle and innermost layers represent the midpapillary and apical segments of the left ventricle, respectively. Red segments correspond to an average T:C ratio of >2; orange segments correspond to an average T:C ratio >1.5–2; yellow segments correspond to an average T:C ratio of >1–1.5; green segments correspond to an average T:C ratio of >0.5–1; blue segments correspond to an average T:C ratio of <0.05. Infant age class includes individuals of ≤4 years of age, juveniles between 5–7 years of age, sub-adults between 8–11 years of age and adults ≥12 years of age.

**Fig. 3 | Comparison of left ventricular morphology and mechanical indices of ventricular function between chimpanzees and humans.**
Shortening along the long axis of the left ventricle (i.e., longitudinal strain) and deformation at the apex (i.e., apical circumferential and apical radial strain) were averaged across a mixed-sex, adult cohort of chimpanzees ($n = 136$) and represented in maroon. Blue reflects the deformation patterns of a mixed-sex, adult human cohort ($n = 34$). Dashed line represents aortic valve closure. Gray shading to the left of dashed line represents systole and white represents diastole.

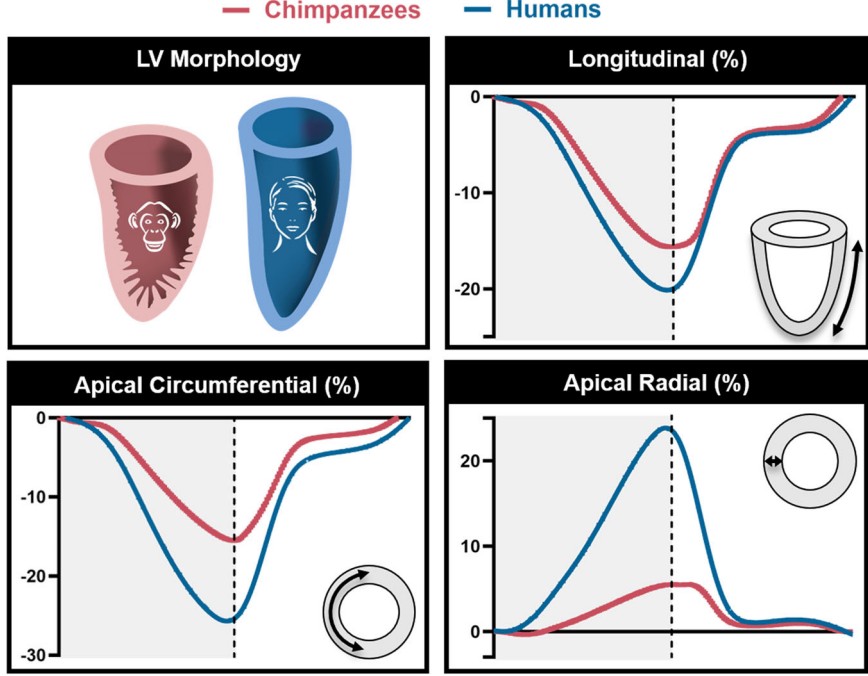

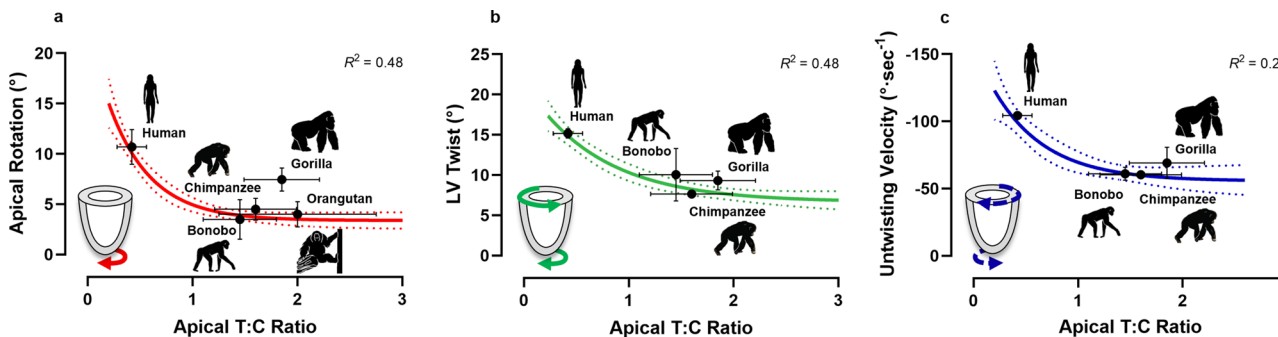

**Fig. 4 | Relationship between markers of left ventricular (LV) function and apical trabeculation in the extant *Hominidae* taxon. a** Peak LV systolic apical rotation, shown in red, in a mixed-sex, adult cohort of humans (male $n = 18$, female $n = 16$), chimpanzees (male $n = 59$, female $n = 51$), bonobos (male $n = 2$, female $n = 4$), gorillas (male $n = 4$, female $n = 6$) and orangutans (male $n = 10$, female $n = 6$). **b** Peak LV systolic twist, shown in green, and (**c**) peak diastolic untwisting velocity, shown in blue, in a cohort of humans (male $n = 18$, female $n = 16$), chimpanzees (male $n = 47$, female $n = 43$), bonobos (male $n = 1$, female $n = 3$) and gorillas (male $n = 4$, female $n = 5$). Analyses of LV twist and untwisting velocity were not possible in all individuals, nor any of the orangutans due to artifacts from laryngeal air sacs, hence the reduced sample size. The exponential plateau curve is shown, with the 95% confidence bands represented by the dotted line. The mean and standard error are shown in black for each species.

$P < 0.001$; two-way ANOVA; Supplementary Table 3). The velocity of ventricular untwisting during diastole and ventricular deformation in the circumferential (i.e., myocardial shortening along the circular perimeter of the LV cavity), radial (i.e., myocardial thickening towards the LV cavity) and longitudinal (i.e., myocardial shortening along the long axis of the LV) planes were also comparatively lower in chimpanzees than humans (all $P < 0.001$; two-way ANOVA; Supplementary Table 3; incorporated into Fig. 3). When male and female chimpanzees were compared, we noted that females had higher circumferential and longitudinal deformation (all $P < 0.001$; two-way ANOVA), and LV twist ($P = 0.017$; two-way ANOVA) in comparison to males.

Overall, these data suggest lower levels of ventricular rotation and deformation in chimpanzees compared with humans. We subsequently explored whether the degree of trabeculation at the LV apex, the region in which the trabeculae appeared most prominent, was associated with ventricular rotation and deformation characteristics. Across all adult great apes, we found a negative curvilinear relationship between the degree of apical trabeculation and (i) LV twist ($R^2 = 0.48$) and (ii) apical rotation

($R^2 = 0.48$), whereby a more trabeculated LV apex was associated with lower systolic twist and rotation (Fig. 4). Furthermore, a positive curvilinear relationship existed between apical trabeculation and the velocity of LV untwisting ($R^2 = 0.29$; Fig. 4), suggesting an association between a more trabeculated LV and slower ventricular untwisting during diastole. No significant relationships were observed between the degree of trabeculation and circumferential, radial or longitudinal deformation characteristics.

## Discussion
The gross morphology of the mammalian heart has remained highly conserved across species[1]. However, the remarkable diversification of this class to accommodate a broad range of ecological niches, and associated hemodynamic conditions, is likely associated with interspecific variation of the cardiac phenotype. Here, we adopted a comparative approach to investigate divergent evolution of the human heart, by comparing LV morphology and mechanical indices of LV systolic and diastolic function across all extant species of the *Hominidae* taxon.

Preliminary work suggested adult male chimpanzees possessed prominent myocardial trabeculations that protrude into the ventricular cavity[3,4]. However, whether this phenotype is also present in other extant non-human great apes, and thus representative of the ancestral phenotype from which humans evolved, remained unknown. In contrast to the comparatively smooth ventricular wall typically observed in healthy humans, we have identified a more trabeculated ventricular myocardium across all non-human great apes using 2D echocardiography. This was particularly apparent at the LV apex, where the difference in the degree of trabeculation between human and non-human great apes was approximately four-fold. These findings underline the distinctiveness of the human ventricular wall, which has a greater proportion of compact to trabecular myocardium, and suggest that a trabeculated LV represents the ancestral phenotype of the *Hominidae* taxon. We then sought to explore the systolic and diastolic functional characteristics of the non-human great ape LV and whether these relate to the trabeculated ventricular wall. We identified lower LV rotation, untwisting velocity, and myocardial deformation compared with humans (Fig. 4; Supplementary Table 3). Additionally, across all species, a curvilinear relationship existed between the degree of trabeculation and the magnitude of systolic LV twist and myocardial rotation at the LV apex, and the velocity of LV untwisting during diastole (Fig. 3). Together, these results suggest divergence of the human LV away from the ancestral phenotype of non-human great apes, providing insight into the evolution of the human heart.

In contrast to their non-human ancestors, *Homo* evolved to occupy a distinct ecological niche in which they engaged in comparatively greater amounts of bipedal locomotion[7,14] and developed larger brains[6]. This increase in daily physical activity and brain size would dictate an increase in metabolism[15] and greater thermoregulatory stress[16], which would have placed selective pressure on the heart to adapt to meet the metabolic and thermoregulatory demands of the human niche. Ex vivo human heart experimentation and modeling studies have indicated that removal of trabeculation improves LV compliance[17] and likely enables a greater LV end-diastolic volume[17,18]. Additionally, divergent evolution of the human LV towards a ventricular wall with proportionately greater compact myocardium may have enhanced myocardial rotation and deformation, and thus aided efficient ventricular filling and ejection[19] due to its potential influence on myocardial architecture. While trabeculated myocardium contains a random distribution of fibers reflecting a meshwork[20,21], compact myocardium is highly-organized[20]. Indeed, the ability of the LV to rotate and deform[20,22–27] is dependent upon the helical angulation of the aggregated caardiomyocytes[9,10]. Interestingly, previous work in macaques (*Macaca nemestrina, Macaca cyclopis*), which share a common ancestor with great apes, has shown a highly trabeculated myocardium coincident with a high degree of disorder in the orientation of the aggregated cardiomyocytes[20]. If this non-uniform structural arrangement is also present in the trabeculated LV in non-human great apes, this may help to explain the lower rotation and deformation observed in these species (Fig. 3). However, further research is needed to specifically determine the myocardial architecture of the LV in non-human great apes, and its relationship with myocardial function.

While a less trabeculated, more compact, ventricular wall might have been favored by natural selection in *Homo*, it remains unclear whether the trabeculae in the non-human great apes are adaptive or if they are a vestigial structure of the ancestral heart. Our data indicate a greater extent of trabeculation at the LV apex, where the myocardium may be at its thinnest and ventricular wall stress may be greatest[28]. Thus, it is possible that the presence of trabeculae could be adaptive and mitigate myocardial wall stress in non-human great apes[17,18]. Additionally, recent work has shown that trabecular cardiomyocytes have a similar contractile force potential to that of compact cardiomyocytes[29] and thus may play a crucial role in the generation of stroke volume in these species. For example, the diastolic expansion and systolic compression of the intertrabecular recesses, created by relaxation and contraction of the trabecular meshwork, might aid LV filling and ejection through the generation of pressure gradients. While we observed a trabeculated ventricular phenotype across all non-human great apes, our data suggest that LV apical trabeculation may be greatest in the cohort of

orangutans. Although based on a small sample, this finding may indicate evolutionary regression in LV trabeculation before the emergence of *Homo*; however, additional data from larger cohorts of orangutans, gorillas, bonobos, and other, more distally related primate species are required to fully elucidate the evolution of this structural phenotype and its relevance to overall cardiac function.

In addition to the interspecific differences we have noted, our data also highlight potential sex-differences in chimpanzee hearts. Males appear to have a greater degree of trabeculation at the LV apex in comparison with females. One possible explanation for this is the known sexually dimorphic pattern of aggression in chimpanzees[30]. Males display much greater rates of aggression than females[31], and so likely experience more frequent surges in blood pressure and, in turn, ventricular wall stress. The greater degree of apical trabeculation in males may therefore help mitigate the hemodynamic stress associated with this behavior. Our results also suggest that female chimpanzees have greater apical rotation, LV twist, and systolic deformation in the circumferential and longitudinal planes compared with males (Supplementary Table 3). These findings align with previous data in humans[32] and may reflect the need for augmented deformation to support the cardiac output of a smaller ventricle size, as has previously been observed in female chimpanzees[33].

Given the genetic proximity of human and non-human great apes, our data may have relevance to the debate surrounding cardiac trabeculation and the concept of left ventricular noncompaction cardiomyopathy[34–37]. Noncompaction cardiomyopathy was originally proposed to result from an aberration of cardiogenesis, whereby the embryonic trabecular myocardium failed to compact[38,39]. However, recent studies refute this notion and indicate that allometric growth of the trabecular and compact myocardium during embryogenesis may explain this morphology[37,40]. Thus, an Expert Panel recently recommended the term "excessive trabeculation", as opposed to the misnomer of "non-compaction"[34]. These authors explain how excessive trabeculation of the ventricular wall is not a clinical entity in itself but may be documented alongside other cardiovascular symptoms or abnormalities[34]. Furthermore, excessive trabeculation can also present as a normal variant or as a physiological response to conditions of altered cardiac load in otherwise healthy humans[41–45]. Similar to humans, albeit to a greater extent, there also appears to be variability in the degree of trabeculation observed in non-human great apes, with some animals showing markedly greater trabeculation than others (see range data in Supplementary Table 1). While our data from non-human great apes highlight that a trabeculated ventricle reflects the normal phenotype in these species, and likely the phenotype from which early hominins diverged, further studies are required to better understand the variability of trabeculation across all great apes; and, whether the presence of excessive trabeculations in some humans might share a genetic and/or physiologic substrate with that of non-human great apes.

Collectively, the findings of this study support evolutionary divergence of the human LV away from a trabeculated ancestral phenotype, towards a ventricular wall with a proportionately greater compact myocardium. We propose that this adaptive evolution occurred to support the requirements of the human ecological niche, including an augmented cardiac output to facilitate sustained bipedal physical activity, a larger brain, and the associated metabolic and thermoregulatory demands.

## Methods
### Sample
Echocardiographic examinations were performed in chimpanzees ($n = 242$, aged $14.8 \pm 7.5$ years, range 1–36 years) housed at one of three Pan African Sanctuary Alliance (PASA) member sanctuaries (Tchimpounga Chimpanzee Rehabilitation Centre, Congo; Chimfunshi Wildlife Orphanage, Zambia; Tacugama Chimpanzee Sanctuary, Sierra Leone) and were cared for in accordance with the PASA operations manual[46]. Information on the husbandry of these animals have been published previously[33,47]. The orangutans ($n = 28$, aged $10.4 \pm 6.8$ years, range: 1–25 years) included in this study were cared for at the Nyaru Menteng Orangutan Rescue and Rehabilitation Center, Borneo, and the gorillas ($n = 14$, aged $24.2 \pm 15.3$ years,

range 4–50 years) and bonobos ($n = 7$, aged 20.0 ± 14.8 years, range 7–46 years) were cared for at European zoological institutions, in accordance with the European Association of Zoos and Aquaria best practice guidelines. As part of the International Primate Heart Project, complete cardiac examinations were performed for all great apes during preplanned health assessments or routine veterinary procedures conducted between 2013 and 2019. Only animals that were non-pregnant and presumed healthy based on physical and echocardiographic examinations were included in the study. The procedures and protocols involved in this study have been approved by, and adhered to, the PASA professional, ethical, and welfare standards, endorsed by the British and Irish Association of Zoos and Aquariums, and ethically approved by the University of British Columbia, Canada (ethics approval number A23-0074). We have complied with all relevant ethical regulations for animal use.

Echocardiographic images of healthy human adult males ($n = 18$, mean age, 23.8 ± 2.8 years; body mass, 75.1 ± 8.5 kg; height, 179.8 ± 6.3 cm) and females ($n = 16$, mean age 22 ± 3.3 years; body mass 62.7 ± 8.3 kg; height 168.4 ± 7.3 cm) were combined from previous studies for the human cohort[48–50]. Importantly, our human data align with those previously reported in large, healthy populations[51]. All individuals were normotensive, non-smokers with no history of cardiovascular, musculoskeletal or metabolic disease. All procedures conformed to the ethical guidelines of the 1975 Declaration of Helsinki, and the studies were approved by the Institutional Clinical Research Ethics Board of the University of British Columbia (ethics approval numbers H12-03531 and H15-01513) and the Cardiff School of Sport and Health Sciences Research Ethics Committee (ethics approval number 17/3/01S). All ethical regulations relevant to human research participants were followed.

### Anesthetic protocols for non-human great apes

Prior to immobilization, all animals were fasted overnight with water available *ad libitum*. Chimpanzees were anesthetized with one of four protocols: (i) combination of medetomidine (0.03–0.05 mg/kg) and ketamine (3–5 mg/kg) delivered intramuscularly via hand injection or remote dart injection ($n = 147$); (ii) combination of tiletamine-zolazepam (2 mg/kg) and medetomidine (0.02 mg/kg) via remote dart injection ($n = 68$); (iii) tiletamine-zolazepam (6 mg/kg) via remote dart injection ($n = 27$). Gorillas were anesthetized with a combination of medetomidine (0.04 mg/kg) and ketamine (4 mg/kg), delivered intramuscularly via remote dart injection. Orangutans received a combination of ketamine (2–3 mg/kg) and xylazine (1 mg/kg), delivered intramuscularly via remote dart injection. Bonobos were anesthetized using a combination of ketamine (2.5–8 mg/kg), xylazine (0.5–0.64 mg/kg), and atropine (0.001–0.004 mg/kg), delivered intramuscularly via remote dart injection. Animals that received medetomidine were reversed with atipamezole (2.5–5 times the dose of medetomidine received, delivered intramuscularly) after completion of the examination. The anesthetic protocol for each examination was determined by the lead veterinarian at each sanctuary or zoological institution, and the dose was based on an estimated body mass based on previous health checks.

### Transthoracic echocardiography protocol

All transthoracic echocardiographic examinations were performed using a commercially available ultrasound machine (Vivid q, GE Vingmed Ultrasound, Horten, Norway and Vivid E9, GE Healthcare, Chalfont St Giles, Bucks, UK) with an appropriate cardiac transducer for the respective age of the individual (infants and juveniles: 6S pediatric cardiac ultrasound transducer, GE Vingmed Ultrasound, Horten, Norway; sub-adults and adults: M5S-D, GE Vingmed Ultrasound, Horten, Norway). All echocardiographic assessments followed a strict protocol developed on the recommendations of the American Society of Echocardiography[52] and the British Society of Echocardiography[53], in conjunction with practical guidance for echocardiography in great apes where applicable[54]. All non-human great ape examinations were performed by a single highly-trained sonographer (A.L.D). Echocardiographic images were acquired from parasternal short-axis and apical four-chamber views, with three cardiac

cycles recorded for offline analysis (EchoPac, GE Medical, Horten, Norway, version 204).

### Assessment of LV trabeculation

In each of our great ape species, LV trabeculation was assessed using the Jenni criteria, an established method to quantify the extent of trabeculation in humans[34,55]. This criterion is based upon the end-systolic ratio of the trabecular to the compact myocardium (T:C ratio), which was assessed using short-axis echocardiographic images of the LV at the levels of the mitral valve, midpapillary, and apex (Supplementary Fig. 2), in accordance with the 16-segment model recommended by the American Society of Echocardiography[56]. The T:C ratio could not be appropriately assessed at the levels of the mitral valve and midpapillary in the orangutans due to substantial artifacts from the laryngeal air sacs, and hence was only available for the apical segments. All analyses were performed by a single researcher (B.A.C). The average intra-observer coefficient of variation for measurement of compact and trabeculated myocardium for all LV segments was 9%. LV trabeculations were also assessed in a small cohort of post-mortem chimpanzee hearts ($n = 15$). Hearts were collected from chimpanzees that died of non-cardiac-related causes and were immediately prepared for fixation in 10% formalin. Transverse slices of ~1 cm were taken starting at the apex, continuing to the apical level of the mitral and bicuspid valve by an experienced veterinarian trained in cardiac pathology (J.E.G), and a macroscopic examination was performed.

### LV rotation and deformation

As age-related differences in LV rotation and deformation have been reported in humans[13], only mature individuals were included in this analysis. Following the exclusion of immature animals, the subsequent sample sizes of the great ape populations were not sufficient for all species to be included in statistical analyses, and hence markers of LV systolic and diastolic function were only compared between adult chimpanzees and humans. Echocardiographic images were exported for 2D speckle-tracking analysis using a commercially available software (EchoPac, GE Medical, Horten, Norway, version 204). Rotation and deformation of the entire myocardium in the circumferential and radial planes (i.e., circumferential and radial strain) were assessed in parasternal short-axis views of the LV base and apex by a single researcher (B.A.C). The LV apex was defined as the point just proximal to end-systolic luminal obliteration[57]. Deformation in the longitudinal plane (i.e., longitudinal strain) was assessed in an apical-four chamber view, and all images were acquired at a frame rate of 60–90 frames per second. LV twist and untwisting velocity were calculated by subtracting short-axis apical data from basal[58]. Frame-by-frame data were exported to bespoke software (2D Strain Analysis Tool; Stuttgart, Germany), and cubic spline interpolation was completed to time-align data. Intra-observer coefficient of variation for LV rotation and deformation within our group has been previously reported to be between 8 and 11%[59]. Appropriate speckle-tracking analysis was not possible in all individuals, hence the reduced sample size of these variables. To confirm that LV rotation and deformation are related to global measures of cardiac function in chimpanzees, as has been reported in humans, we explored the relationship between ejection fraction, and LV longitudinal strain and LV twist (Supplementary Fig. 3).

When interpreting our results, it is important to acknowledge the potential influence of anesthesia on myocardial function. Data in the non-human great apes were collected across several institutions, all of which adopt different anesthetic protocols. Dependent on anesthetic agent, these protocols may result in periods of either hypo- or hypertension[60]. LV rotation and deformation, and the rate at which it occurs, can be influenced by changes in blood pressure[61,62]. Despite carefully aligning data collection with the most hemodynamically stable period of anesthesia, as has been previously recommended[63], it is possible that our functional data may have been somewhat influenced. However, there was no overlap of the sex-specific 95% confidence intervals for our primary variables of interest (i.e., LV twist, untwisting velocity, apical rotation, apical radial strain, apical

circumferential strain, and longitudinal strain) between the human and chimpanzee cohort. Therefore, the difference in the peak rotation and deformation values between the chimpanzee and human cohorts was generally large – beyond that which would be expected through modest hemodynamic changes with anesthesia. In support of this, the only study to examine LV deformation in awake (non-anesthetized) great-apes reported a mean LV longitudinal strain of $-16.3 \pm 0.7\%$ in $n = 4$ orangutans[64], which is very similar to our chimpanzee data (males: $-15.0 \pm 2.2\%$; females $-19.0 \pm 2.4\%$).

## Statistics and reproducibility
Data were first assessed for normality and equality of variances using the Shapiro-Wilk test and Levene's test, respectively. The average T:C ratio was compared for each LV segment across great ape species using multiple one-way independent Kruskal-Wallis tests, with Dunn-Bonferroni post hoc analyses. To further assess whether the average T:C ratio for each LV segment differed with age and/or sex in the chimpanzee cohort, a two-way analysis of variance (ANOVA) was used with Bonferroni post hoc analyses, for which age group and sex were independent variables. Additionally, LV rotation and deformation were compared within sex and between adult chimpanzees and humans using a two-way ANOVA with Bonferroni post hoc analyses. Alpha was set at $P < 0.05$. All statistical analyses were performed using the Statistical Package for the Social Sciences version 28 (SPSS Inc.). The relationships between the apical T:C ratio and LV rotation and deformation parameters were subsequently explored using exponential plateau curves, which were chosen based on the predicted relationship (i.e., a non-linear relationship was expected between our variables of interest as preliminary data[3] indicated some degree of rotation and trabeculation) and fit of the model (GraphPad Prism version 9.4.1 for Windows; San Diego, CA). Apical rotation and deformation parameters were selected for these analyses as this was the most trabeculated level of the LV observed in the non-human great ape cohort (Supplementary Table 1). However, apical rotation, LV twist, and untwisting velocity were the only parameters to demonstrate a significant relationship with the average apical T:C ratio.

## Reporting summary
Further information on research design is available in the Nature Portfolio Reporting Summary linked to this article.

## Data availability
The data that support the findings of this study are available in Figshare repository with the identifiers: https://doi.org/10.6084/m9.figshare.24274852 and https://doi.org/10.6084/m9.figshare.24274855.

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

## Acknowledgements
We thank all the staff and volunteers that care for the animals included in this study, particularly the teams at Tchimpounga Wildlife Sanctuary (Congo), Chimfunshi Wildlife Sanctuary (Zambia), Tacugama Chimpanzee Sanctuary (Sierra Leone), Nyaru Menteng Orangutan Rescue and Rehabilitation Center (Borneo), the Zoological Society of London (UK), Paignton Zoo (UK), Bristol Zoo Gardens (UK), Burgers' Zoo (Netherlands) and Wilhelma Zoo (Germany). This work was supported by the Natural Sciences and Engineering Research Council, grant no. GR017741 (R.E.S.), and the Canadian Foundation for Innovation grant no. GR014935 (R.E.S.).

## Author contributions

B.A.C., A.L.D and R.E.S designed research; B.A.C., A.L.D., R.A., Y.F., T.C., E.L.M., S.M., A.W., T.K.-W., A.S.K, G.H., C.P., M.R.S., J.E.G., N.D.E, T.G.D and R.E.S. performed research; B.A.C and A.L.D analyzed data; B.A.C., A.L.D., T.G.D. and R.E.S wrote the paper.

## Competing interests

The authors declare no competing interests.

## Additional information

[1]Centre for Heart, Lung and Vascular Health, School of Health and Exercise Sciences, University of British Columbia, Kelowna, BC V1V 1V7, Canada. [2]International Primate Heart Project, Cardiff Metropolitan University, Cyncoed Road, Cardiff CF23 6XD, UK. [3]Faculty of Medicine, Health and Life Sciences, Swansea University, Swansea SA2 8PP, UK. [4]Jane Goodall Institute, Tchimpounga Chimpanzee Rehabilitation Centre, Pointe-Noire, Republic of Congo. [5]Chimfunshi Wildlife Orphanage, Solwesi Road, Chingola, Zambia. [6]Wildlife Health, Pathobiology and Population Sciences, Royal Veterinary College, University of London, Hawkshead Lane, North Mymms, Hatfield, Hertfordshire AL9 7TA, UK. [7]Zoological Society of London, Regent's Park, London NW1 4RY, UK. [8]Centre for Veterinary Wildlife Research, Faculty of Veterinary Science, University of Pretoria, Private Bag X04, Onderstepoort, Pretoria 0110, South Africa. [9]Tacugama Chimpanzee Sanctuary, Congo Dam Access Road, Freetown, Sierra Leone. [10]School of Veterinary Medicine, St. George's University, St. George's, West Indies, Grenada. [11]Wilhelma Zoological-Botanical Gardens, Wilhelma 13, Stuttgart 70376, Germany. [12]Borneo Orangutan Survival Foundation, Central Kalimantan Orangutan Reintroduction Project at Nyaru Menteng, Jalan Cilik Riwut km 28, Palangkaraya 73111 Central Kalimantan, Indonesia. [13]Faculty of Health and Life Sciences, Northumbria University, Newcastle-upon-Tyne NE1 8ST, UK. [14]Water Research Group, Faculty of Natural and Environmental Sciences, North West University, Potchefstroom 2531, South Africa. [15]Biological Science, University of New South Wales, Sydney, NSW 2052, Australia. [16]Cardiff School of Sport and Health Sciences, Cardiff Metropolitan University, Cardiff CF23 6XD, UK. [17]Department of Cardiovascular Medicine, Stanford University School of Medicine, Stanford, CA 94305, USA. ✉e-mail: a.l.drane@swansea.ac.uk; rob.shave@ubc.ca

