## [Peer Review File · Communications Biology]

Reviewers' comments:

Reviewer #1 (Remarks to the Author):

You have explored a fascinating topic, and one that is becoming of increasing importance relative to human cardiac disease. It is encouraging that, throughout the manuscript, you use the phrase "excessive trabeculation". If your hypothesis is correct, however, then surely the extent of trabeculation that you have described and illustrated, in the species you have investigated, is far from excessive? If you are correct, the extent of trabeculation is the norm for all of the species. It is only excessive when compared with the arrangement considered to be the "norm" for the human heart. In your references, late in your account, you cite the recent work of Petersen and colleagues where they discuss the overall topic of excessive trabeculation relative to so-called "non-compaction". Nowhere in your otherwise authoritative account, however, do you discuss this potentially controversial issue. In others of your cited references, nonetheless, it is clear that the cited authors are describing so-called "non-compaction". This is particularly pertinent when you suggest that "excessive trabeculation" is to be found in neonates. The cited reference, however, is reporting the findings in cardiomyopathic neonates, with your next cited reference similarly reporting findings in abnormal, rather than normal, human hearts. You are not, therefore, comparing like with like. Your potentially important findings are incomplete without a discussion also of the issues of "non-compaction" versus excessive trabeculation. It is likely that a proportion of otherwise normal humans have excessive trabeculation. Findings from the UK Biobank study support this notion. It is also the case that you cite a reference from Jensen that is well out of date. Jensen and his colleagues have subsequently published several articles arguing that trabeculated myocardium can be just as forceful as compact myocardium. Should not attention also have been directed at this possibility? You then place significant emphasis on the concept of "ventricular twist". This is also, however, a contentious topic. Sengupta and his colleagues, who have written one of your cited references, consider the "twist" to be the consequence of the presence, within the human ventricular cone, of a "unique myocardial band". There is no evidence to support this notion, but there is considerable evidence to show that it is the ventricular mural architecture that underscores the twisting motion of the ventricular walls. If you are considering the changes you have observed to be the consequence of the excessive trabeculation, should you not also establish that the architecture of the cardiomyocytes aggregated to form the compact wall of the great apes are comparable to the arrangement known to exist in the human heart? One of your cited references, again, discusses this feature (the work of Robb and Streeter), yet you do not subsequently discuss this potentially conflicting factor. In summary, your findings are of potential importance, but should also be discussed in the context of so-called "non-compaction", and also relative to ventricular mural architecture. You should also establish whether, as in humans, there is variation in the extent of the ratios of the trabeculated versus compact components of the ventricular walls amongst your apes. Note also that, in panel B of the images included in your supplementary material, there seems to be an apical ventricular septal defect in the chimpanzee heart illustrated. It is less than ideal to use an image that itself raises additional questions. Panel A is sufficiently convincing in its own right.

Reviewer #2 (Remarks to the Author):

The manuscript by Curry et al is based on the highly interesting research question, whether the human heart - but actually left ventricle rather than whole heart - exhibits overt structural adaptations to sustain the prolonged periods of high physical activity that sets human apart from its nearest relatives. A great many chimpanzees (n = 242), bonobos (n = 7), gorillas (n = 14), and orangutans (n = 28) have been investigated with echocardiography. A compelling case is made that the trabeculations of the human left ventricle are less prominent than in the other great apes. Functional indices of strain and rotation of the left ventricle indicate higher functionality of the human left ventricle.

I have the following major comments;

1. This study reanalyzes and expands on the findings of Shave et al 2019 PNAS. Much of the data is from previous studies. One could question how much new we learn from the present manuscript; we would already presume the primitive condition relative to the human setting likely was a quite trabeculated left ventricle. The most exciting new data is on the orangutans, but this data is only partially complete due to unsurmountable technical challenges.

2. The data on the orangutans and gorillas is suggestive of even greater LV trabeculation than in the chimpanzees, bonobos, and human. Does this not suggest that there was an evolutionary regression of left ventricular trabeculation before Homo?

3. That less trabeculation could be an adaptation to (episodes of) greater cardiac output, is not obviously supported by a) the right ventricle is much more trabeculated than the left ventricle but delivers exactly the same cardiac output, and b) the findings in cohort studies in human where individuals with relatively high activity levels tend to have more prominent trabeculations (PMID: 30723101; PMID: 33032733). Can a cogent argument be made how less trabeculation could favor/facilitate greater cardiac output?

4. In the Discussion, citing reference 19, you write that the trabecular layer “may assist in LV filling and ejection by ‘sucking’ and ‘squeezing’ blood”. It is seemingly implied here but actually in the whole manuscript that trabecular myocardium is less functional than compact myocardium and this immediately raises the conundrum of why is there any trabeculation at all? The two types of myocardium, however, are highly similar on tissue and cellular level and in contractile strength (PMID: 36345331). Thus, your more or less explicit presumption that compact myocardium is more functional than trabecular myocardium is likely untenable.

5. Regarding LV wall rotation and strain, these analyses are restricted to the compact wall. That is, the more trabeculated the wall is, the smaller the fraction of total myocardium is analyzed, which begs the question whether such analyses are biased towards lower readouts at higher degrees of trabeculation? In addition, the trabecular layer is a meshwork, that is, any position in it has two or multiple orientations, or, in aggregate, no overall orientation. It is then improbable that the metrics of the compact wall applies equally well to the trabecular layer. In this setting, the absence of good readouts in the trabecular layer, does not mean that good readouts/function cannot exist in the trabecular layer.

6. The functional analyses suggest the non-human great ape left ventricles are borderline failing, which makes little sense. Rather, one wonders whether the differences in functional readout is a consequence

of the chimpanzees experiencing a higher afterload (due to anesthetics) than the non-sedated humans? How do your findings compare to those on awake orangutans (PMID: 35073314)? To clarify the importance of this potential confounding issue, it would be valuable to see the echo readouts on stroke volume and ejection fraction.

7. What was the rationale - statistical and biological - for assuming a curvilinear relation rather than linear (fewer degrees of freedom) relation between trabecular-to-compact ratio and functional indices?

Response to Reviewers

Your manuscript entitled "Left ventricular trabeculation in Hominidae: divergence of the human cardiac phenotype" has now been seen by 2 referees, whose comments are appended below. You will see from their comments copied below that while they find your work of potential interest, they have raised quite substantial concerns that must be addressed. In light of these comments, we cannot accept the manuscript for publication, but would be interested in considering a revised version that addresses these serious concerns.

In particular, please note that the following revisions would be necessary for us to contact our referees again: please carefully address all concerns raised by the reviewers, with particular attention to those related to the contextualization and interpretation of your findings, which would require substantial revisions in the introduction and discussion sections of the manuscript.

We are grateful for the helpful suggestions from the reviewers and Senior Editor. Based on these, we have made extensive revisions to our discussion to specifically address the contextualization and interpretation of our findings. In addition, within the introduction, we now introduce the importance of the helical arrangement of the LV fibers so as to set up a clearer line of argument with regard to the potential evolutionary advantages associated with a proportionately greater compact myocardium in *Homo*.

Reviewer #1

1. You have explored a fascinating topic, and one that is becoming of increasing importance relative to human cardiac disease. It is encouraging that, throughout the manuscript, you use the phrase "excessive trabeculation". If your hypothesis is correct, however, then surely the extent of trabeculation that you have described and illustrated, in the species you have investigated, is far from excessive? If you are correct, the extent of trabeculation is the norm for all of the species. It is only excessive when compared with the arrangement considered to be the "norm" for the human heart.

We thank the reviewer for their very positive comments regarding our manuscript, and this helpful comment. We fully agree with the reviewer's suggestion that trabeculation in the great ape left ventricle should only be considered 'excessive' in relation to the 'norm' for the human heart. We have amended the text in the manuscript to reflect this by removing the phrase 'excessive trabeculation' from the introduction of the original manuscript (on page 4). In line with the reviewer's comments, we have referred to this term in the revised manuscript in the discussion of excessive trabeculation versus left ventricular noncompaction (lines 242 - 258).

2. In your references, late in your account, you cite the recent work of Petersen and colleagues where they discuss the overall topic of excessive trabeculation relative to so-called "non-compaction". Nowhere in your otherwise authoritative account, however, do you discuss this potentially controversial issue...Your potentially important findings are incomplete without a discussion also of the issues of "non-compaction" versus excessive trabeculation. It is likely that a proportion of otherwise normal humans have excessive trabeculation. Findings from the UK Biobank study support this notion.

We thank the reviewer for this helpful comment and appreciate their recognition of the importance of our findings. Previously, we had chosen not to conflate the debate regarding left ventricular non-compaction versus excessive trabeculation in humans with the potential evolutionary regression of a trabeculated ventricle. However, on reflection, we agree with the reviewer that it would be inappropriate not to address the topic of "non-compaction cardiomyopathy" versus excessive trabeculation in humans, and to discuss our findings within this context. In line with this, and the recommendation of the Senior Editor, we have added the following paragraph to our discussion in lines 242 - 258:

'Given the genetic proximity of human and non-human great apes, our data may have relevance to the debate surrounding cardiac trabeculation and the concept of left ventricular noncompaction cardiomyopathy¹⁻⁴. Noncompaction cardiomyopathy was originally proposed to result from an aberration of cardiogenesis, whereby the embryonic trabecular myocardium failed to compact^{5,6}. However, recent studies refute this notion, and indicate that allometric growth of the trabecular and compact myocardium during embryogenesis may explain this morphology^{4,7}. Thus, an Expert Panel recently recommended the term 'excessive trabeculation', as opposed to the misnomer of 'non-compaction'¹. These authors explain how excessive trabeculation of the ventricular wall is not a clinical entity in itself, but may be documented alongside other cardiovascular symptoms or abnormalities¹. Furthermore, excessive trabeculation can also present as a normal variant or as a physiological response to conditions of altered cardiac load in otherwise healthy humans⁸⁻¹². Similar to humans, albeit to a greater extent, there also appears to be variability in the degree of trabeculation observed in non-human great apes, with some animals showing markedly greater trabeculation than others (see range data in Supplementary Table 1). While our data from non-human great apes highlight that a trabeculated ventricle reflects the normal phenotype in these species, and likely the phenotype from which early hominins diverged, further studies are required to better understand the variability of trabeculation across all great apes; and, whether the presence of excessive trabeculations in some humans might share a genetic and/or physiologic substrate with that of non-human great apes.'

3. In others of your cited references, nonetheless, it is clear that the cited authors are describing so-called "non-compaction". This is particularly pertinent when you suggest that "excessive trabeculation" is to be found in neonates. The cited reference, however, is reporting the findings in cardiomyopathic neonates, with your next cited reference similarly reporting findings in abnormal, rather than normal, human hearts. You are not, therefore, comparing like with like.

We thank the reviewer for this comment, and agree that the cited reference was inappropriate within this context. We have now amended the text to highlight that previous work had only considered adult male chimpanzees, and hence an exploration of potential age and sex differences are warranted to determine the ubiquity of a trabeculated phenotype in this species (line 137).

'As previous work only examined adult male chimpanzees¹³, we further investigated whether LV trabeculation varied across age and sex in our large cohort of chimpanzees to determine the ubiquity of this phenotype.'

4. It is also the case that you cite a reference from Jensen that is well out of date. Jensen and his colleagues have subsequently published several articles arguing that trabeculated myocardium can be just as forceful as compact myocardium. Should not attention also have been directed at this possibility?

The reviewer raises an important consideration that was missing from the original manuscript. We thank the reviewer for this comment and have amended the discussion to include this in lines 220 - 224:

'Additionally, recent work has shown that trabecular cardiomyocytes have a similar contractile force potential to that of compact cardiomyocytes¹⁴ and thus may play a crucial role in the generation of stroke volume in these species. For example, the diastolic expansion and systolic compression of the intertrabecular recesses, created by relaxation and contraction of the trabecular meshwork, might aid LV filling and ejection through the generation of pressure gradients.'

5. You then place significant emphasis on the concept of "ventricular twist". This is also, however, a contentious topic. Sengupta and his colleagues, who have written one of your cited references, consider the "twist" to be the consequence of the presence, within the human ventricular cone, of a "unique myocardial band". There is no evidence to support this notion, but there is considerable evidence to show that it is the ventricular mural architecture that underscores the twisting motion of the ventricular walls. If you are considering the changes you have observed to be the consequence of the excessive trabeculation, should you not also establish that the architecture of the cardiomyocytes aggregated to form the compact wall of the great apes are comparable to the arrangement known to exist in the human heart? One of your cited references, again, discusses this feature (the work of Robb and Streeter), yet you do not subsequently discuss this potentially conflicting factor.

We are grateful for the reviewer's suggestion and agree that a greater discussion of the myocardial architecture of the left ventricle is required to explain a possible link between trabeculation and left ventricular twist, rotation and deformation. We have removed the earlier reference that speaks to the outdated "myocardial band" theory and focus on the importance of the myocardial architecture. Unfortunately, no study to date has examined the myocardial architecture of the non-human great ape left ventricle. Ross and Streeter (1975) have examined myofiber orientation of the left ventricle in two rhesus macaques (*Macaca mulatta*) and reported a high degree of disorder in the fiber orientation of the trabeculated myocardium¹⁵. We believe it is, therefore, possible that the trabeculated myocardium of great apes has a similar fiber orientation to that of macaques. As the helical arrangement of the left ventricle, and its associated myofiber angle, are crucial in facilitating myocardial rotation and deformation, it is rational to suggest that potential differences in fiber orientation may explain the observed differences in rotation and deformation between our human and chimpanzee cohorts. However, we acknowledge that further study is required to specifically test this hypothesis. We have added considerably to our discussion to better explain the potential link between trabeculations, fiber architecture and LV rotation and deformation in lines 202 - 214:

*'Additionally, divergent evolution of the human LV towards a ventricular wall with proportionately greater compact myocardium may have enhanced myocardial rotation and deformation, and thus aided efficient ventricular filling and ejection¹⁹ due to its potential influence on myocardial architecture. While trabeculated myocardium contains a random distribution of fibers reflecting a meshwork^{20,21}, compact myocardium is highly-organized²⁰. Indeed, the ability of the LV to rotate and deform^{20,22-27} is dependent upon its helical myofiber arrangement, with the degree of rotation and deformation being influenced by the specific myofiber helix angle^{9,10}. Interestingly, previous work in macaques (*Macaca nemestrina*, *Macaca cyclopis*), which share a common ancestor with great apes, has shown a highly trabeculated myocardium coincident with a high degree of disorder in the fiber orientation²⁰. If this non-uniform structural arrangement is also present in the trabeculated LV in non-human great apes, this may help to explain the lower rotation and deformation observed in these species (Figure 3). However, further research is needed to specifically determine the myocardial architecture of the LV in non-human great apes, and its relationship with myocardial function.'*

6. You should also establish whether, as in humans, there is variation in the extent of the ratios of the trabeculated versus compact components of the ventricular walls amongst your apes.

We thank the reviewer for their comment and have provided the range of left ventricular trabeculation (i.e. the trabecular:compact ratio) for the levels of the mitral valve, midpapillary and apex of each species in Supplementary Table 1. We have included the following in the discussion in lines 253 - 260:

‘Similar to humans, albeit to a greater extent, there also appears to be variability in the degree of trabeculation observed in non-human great apes, with some animals showing markedly greater trabeculation than others (see range data in Supplementary Table 1). While our data from non-human great apes highlight that a trabeculated ventricle reflects the normal phenotype in these species, and likely the phenotype from which early hominins diverged, further studies are required to better understand the variability of trabeculation across all great apes; and, whether the presence of excessive trabeculations in some humans might share a genetic and/or physiologic substrate with that of non-human great apes.’

7. Note also that, in panel B of the images included in your supplementary material, there seems to be an apical ventricular septal defect in the chimpanzee heart illustrated. It is less than ideal to use an image that itself raises additional questions. Panel A is sufficiently convincing in its own right.

We are grateful for the reviewer’s comment, and have amended the SI figure accordingly so that it only contains Panel A.

Reviewer #2

The manuscript by Curry et al is based on the highly interesting research question, whether the human heart - but actually left ventricle rather than whole heart - exhibits overt structural adaptations to sustain the prolonged periods of high physical activity that sets human apart from its nearest relatives. A great many chimpanzees (n = 242), bonobos (n = 7), gorillas (n = 14), and orangutans (n = 28) have been investigated with echocardiography. A compelling case is made that the trabeculations of the human left ventricle are less prominent than in the other great apes. Functional indices of strain and rotation of the left ventricle indicate higher functionality of the human left ventricle.

We thank the reviewer for their comments and critique of our study. We have addressed each comment to provide greater clarity, and where possible, have made significant amendments to the manuscript. We believe these have significantly improved the manuscript, and we hope the changes we have made are in line with the reviewer’s suggestions.

I have the following major comments;

1. This study reanalyzes and expands on the findings of Shave et al 2019 PNAS. Much of the data is from previous studies. One could question how much new we learn from the present manuscript; we would already presume the primitive condition relative to the human setting likely was a quite trabeculated left

ventricle. The most exciting new data is on the orangutans, but this data is only partially complete due to unsurmountable technical challenges.

We thank the reviewer for their comment, however we strongly believe the present manuscript is a valuable contribution to the literature on human evolution and left ventricular trabeculation. The reviewer is correct that previous work from our group did indeed identify ventricular trabeculation in non-human great apes. However, that study was conducted using a small cohort of adult male chimpanzees ($n=43$), and a gorilla outgroup (which was extremely limited in number; $n=5$). Comparative physiology is fraught with assumptions based on inadequate data. It would be inappropriate to ‘presume’ that the primitive condition relative to humans is a trabeculated ventricle based on our previous work. Rather than a reanalysis of previous data, the present manuscript analyzes cardiac data across all extant species of the *Hominidae* taxon. Furthermore, because of the inclusion of 242 chimpanzees, we have also been able to statistically assess whether age and/or sex influences the ratio of trabecular to compact myocardium and have therefore identified that this is a ubiquitous phenotype of chimpanzees. As such, the present manuscript provides strong evidence for the divergence of the human left ventricle away from a trabeculated ancestral phenotype. We therefore believe this manuscript is a beneficial addition to the current literature on human evolution and cardiac trabeculation.

2. The data on the orangutans and gorillas is suggestive of even greater LV trabeculation than in the chimpanzees, bonobos, and human. Does this not suggest that there was an evolutionary regression of left ventricular trabeculation before *Homo*?

We agree with the reviewer that our data from the orangutan and gorilla cohorts may be suggestive of an earlier evolutionary regression, and we respectfully refer them to the discussion of the original manuscript (at the end of page 9), where we specifically raised this possibility. In the revised version of the manuscript, we once again refer to this point in lines 224 - 230:

*‘While we observed a trabeculated ventricular phenotype across all non-human great apes, our data suggest that LV apical trabeculation may be greatest in the cohort of orangutans. Although based on a small sample, this finding may indicate evolutionary regression in LV trabeculation before the emergence of *Homo*; however, additional data from larger cohorts of orangutans, gorillas, bonobos and other, more distally related primate species are required to fully elucidate the evolution of this structural phenotype and its relevance to overall cardiac function.’*

Although this is an interesting concept, we think it important to emphasize that without greater data we need to be cautious not to overinterpret these findings – hence, the addition of the caveat regarding sample size in our text.

3. That less trabeculation could be an adaptation to (episodes of) greater cardiac output, is not obviously supported by a) the right ventricle is much more trabeculated than the left ventricle but delivers exactly the same cardiac output, and b) the findings in cohort studies in human where individuals with relatively high activity levels tend to have more prominent trabeculations (PMID: 30723101; PMID: 33032733). Can a cogent argument be made how less trabeculation could favor/facilitate greater cardiac output?

We thank the reviewer for raising these points that have helped us refine, and clarify, our argument. In response to the reviewer's point (a), we are not convinced that a comparison with the right ventricle (RV) is overly informative. It is important to consider the distinct differences that exist between the right and left ventricles (LV) and specifically the nature of the circuits into which they eject. As the reviewer is aware, the relatively low pressure within the pulmonary circuit is vastly different from systemic pressure, meaning a straight comparison between the LV and RV is inappropriate. Despite the requirement to deliver the same cardiac output, the LV has been selected to meet the metabolic and thermoregulatory demands at rest and during exercise (i.e. there is a need to generate significantly higher pressures). Whereas, selection on the RV has been for the maintenance of consistent pulmonary blood flow to optimize gaseous exchange at the lung – requiring comparatively lower pressure generation than the LV due to the marked differences in systemic versus pulmonary vascular resistance. Thus, it is not surprising that there are marked structural (and indeed functional) differences between the LV and RV.

In answer to part (b) of the reviewer's comment, we do agree that some studies have shown that individuals with high activity levels may have more prominent trabeculations (PMID: 33032733). Importantly, however, this finding is not consistent in the general population (PMID: 30723101) and has most often been shown in groups of extremely highly trained athletes (e.g. PMID: 23393084). The level of exercise training undertaken by these individuals far exceeds that which humans were likely selected for, and as such, care needs to be taken when drawing comparisons with this cohort of individuals. The phenotype of highly trained human athletes represents a combination of the ancestral phenotype (selected for during evolution), superimposed with the acute adaptations to a supra-normal physiologic stimulus. We respectfully argue therefore that elite athletes are not an appropriate model for interspecific comparisons.

Through careful revision of our manuscript, and based on the insightful suggestions of both reviewers, we believe that we have presented a more cogent argument as to why less trabeculations may support an enhanced cardiac output in the derived human.

4. In the Discussion, citing reference 19, you write that the trabecular layer “may assist in LV filling and ejection by ‘sucking’ and ‘squeezing’ blood”. It is seemingly implied here but actually in the whole manuscript that trabecular myocardium is less functional than compact myocardium and this immediately raises the conundrum of why is there any trabeculation at all? The two types of myocardium, however, are

highly similar on tissue and cellular level and in contractile strength (PMID: 36345331). Thus, your more or less explicit presumption that compact myocardium is more functional than trabecular myocardium is likely untenable.

We thank the reviewer for this important comment, which has prompted us to reflect on the present manuscript. We apologize if the content of the original manuscript suggested that the trabecular myocardium has comparatively less contractile strength than that of compact myocardium – this was not our intention at all and we have amended our discussion accordingly in lines 220 - 224:

‘Additionally, recent work has shown that trabecular cardiomyocytes have a similar contractile force potential to that of compact cardiomyocytes¹⁴ and thus may play a crucial role in the generation of stroke volume in these species. For example, the diastolic expansion and systolic compression of the intertrabecular recesses, created by relaxation and contraction of the trabecular meshwork, might aid LV filling and ejection through the generation of pressure gradients.’

The data from the present paper suggest greater trabeculation is associated with lower left ventricular twist/untwisting velocity/apical rotation in great apes. While we agree with the reviewer that there is no difference in the contractile strength between trabecular and compact myocardium at a cellular level, it is possible that the trabecular myocardium in the non-human great ape LV is associated with a differential myocardial architecture to that of humans (i.e. differences exist at a tissue level). The helical myofiber arrangement, and the associated myofiber angle, are crucial in facilitating LV twist and myocardial deformation. The trabecular meshwork in great apes may be associated with a different fiber angle to that of the endocardium in humans (as was previously noted in macaques¹⁵); however, we fully acknowledge that further study is required to elucidate the myofiber architecture of non-human great apes. We have amended our discussion to reflect this potential explanation in lines 202 – 214:

*‘Additionally, divergent evolution of the human LV towards a ventricular wall with proportionately greater compact myocardium may have enhanced myocardial rotation and deformation, and thus aided efficient ventricular filling and ejection¹⁹ due to its potential influence on myocardial architecture. While trabeculated myocardium contains a random distribution of fibers reflecting a meshwork^{20,21}, compact myocardium is highly-organized²⁰. Indeed, the ability of the LV to rotate and deform^{20,22–27} is dependent upon its helical myofiber arrangement, with the degree of rotation and deformation being influenced by the specific myofiber helix angle^{9,10}. Interestingly, previous work in macaques (*Macaca nemestrina*, *Macaca cyclopis*), which share a common ancestor with great apes, has shown a highly trabeculated myocardium coincident with a high degree of disorder in the fiber orientation²⁰. If this non-uniform structural arrangement is also present in the trabeculated LV in non-human great apes, this may help to explain the lower rotation and*

deformation observed in these species (Figure 3). However, further research is needed to specifically determine the myocardial architecture of the LV in non-human great apes, and its relationship with myocardial function.'

5. Regarding LV wall rotation and strain, these analyses are restricted to the compact wall. That is, the more trabeculated the wall is, the smaller the fraction of total myocardium is analyzed, which begs the question whether such analyses are biased towards lower readouts at higher degrees of trabeculation? In addition, the trabecular layer is a meshwork, any position in it has two or multiple orientations, or, in aggregate, no overall orientation. It is improbable that the metrics of the compact wall applies equally well to the trabecular layer. In this setting, the absence of good readouts in the trabecular layer, does not mean that good readouts/function cannot exist in the trabecular layer.

It is clear that the reviewer has an in-depth understanding of the left ventricular (LV) mechanics data presented in the manuscript, which we are grateful for. The LV strain and rotation data in our paper incorporates both the trabeculated layer and the compact layer within the region of interest. We appreciate the nuances in this approach; the reviewer is correct that conventional analyses do not typically incorporate the trabeculated layer. However, as the reviewer has pointed out, the trabeculated layer has contractile properties and reflects a significant portion of the myocardium in non-human great apes. Therefore, we believed that warranted its inclusion within the region of interest for the analysis, and have amended the methods to better reflect this in lines 339 - 341:

'Rotation and deformation of the entire myocardium in the circumferential and radial planes (i.e. circumferential and radial strain) were assessed in parasternal short-axis views.'

To specifically examine the impact of including versus excluding the trabeculated layer within the region of interest (ROI) for the LV mechanics analyses, we have subsequently performed a separate analysis in a cohort of 11 adult chimpanzees. Data from this analysis is provided below; irrespective of whether the trabeculated layer was included or not, the difference in LV deformation between chimpanzees and humans is still present.

Variable	Chimpanzee sub-cohort Entire myocardium ROI (%)	Chimpanzee sub-cohort Compact myocardium ROI (%)	Male human cohort (%)
Apical rotation	3.30 ± 1.93	2.84 ± 1.52	10.00 ± 3.6
Longitudinal strain	-15.35 ± 2.59	-13.98 ± 1.8	-18.8 ± 1.9

6. The functional analyses suggest the non-human great ape left ventricles are borderline failing, which makes little sense. Rather, one wonders whether the differences in functional readout is a consequence of

the chimpanzees experiencing a higher afterload (due to anesthetics) than the non-sedated humans? How do your findings compare to those on awake orangutans (PMID: 35073314)? To clarify the importance of this potential confounding issue, it would be valuable to see the echo readouts on stroke volume and ejection fraction.

We thank the reviewer for their comment and while we appreciate the reviewers concern, we strongly disagree that our data indicate a borderline failing ventricle. Although very minimal literature has examined left ventricular (LV) mechanics in non-human great apes, and therefore normal ranges for LV mechanics do not currently exist in these species, our data ($n = 132$ chimpanzees) is highly comparable with the orangutan study referenced by the reviewer (PMID: 35073314) which was specifically noted in the original manuscript (on page 14), and remains in the revised manuscript in lines 361 - 363:

'The only study to examine LV deformation in awake (non-anesthetized) great-apes reported a mean LV longitudinal strain of $-16.3 \pm 0.7\%$ in $n = 4$ orangutans⁵⁶, which is very similar to our chimpanzee data (males: $-15.0 \pm 2.2\%$; females $-19.0 \pm 2.4\%$).'

We hope the reviewer will agree that the similarity of our data with that presented in non-sedated orangutans do not support the suggestion of a borderline failing ventricle, but rather that the mechanics of the LV are different between human and non-human great apes. Notwithstanding, we fully acknowledge the potential influence of anesthesia on myocardial function and explicitly discussed this in the methods section of the original manuscript (on page 14) and which remains in the revised manuscript (lines 349 - 363). We do not think, however, that the influence of anesthesia negates the value of the functional data presented in the manuscript. All data were collected during the most hemodynamically stable period of anesthesia²⁶, and the differences in our functional data between the human and chimpanzee cohorts were generally large – well-beyond that which would be expected through modest hemodynamic changes with anesthesia. As noted by the reviewer, it is also informative to examine other cardiac parameters in this context. These parameters (ejection fraction and stroke volume) have been previously reported by our group (Drane et al., 2019: PMID 31140849) and so it would be inappropriate to publish these data twice; however, we note that these data are extremely similar to those reported in the non-sedated orangutans referred to by the reviewer (chimpanzee EF $\sim 55\%$ (dependent upon age)²⁷ vs. $\sim 52\%$ in the small group of awake orangutans). This similarity provides additional support that the influence of anesthesia may not be as critical as has been previously suggested (at least when you pay careful attention to when the measures are made in relation to the time of anesthetic induction).

7. What was the rationale - statistical and biological - for assuming a curvilinear relation rather than linear (fewer degrees of freedom) relation between trabecular-to-compact ratio and functional indices?

We thank the reviewer for their comment. We respectfully refer them to the methods in the original manuscript (page 15), where we referred to this point. However, for further clarification, we have expanded upon our rationale in the revised manuscript in lines 373 - 377:

'The relationships between the apical T:C ratio and LV rotation and deformation parameters were subsequently explored using exponential plateau curves, which were chosen based on the predicted relationship (i.e. a non-linear relationship was expected between our variables of interest as preliminary data¹³ indicated some degree of rotation and trabeculation) and fit of the model.'

Previous work by Ross and Streeter¹⁵ suggested a double helical myofiber arrangement in the left ventricle (LV) of rhesus macaques, which are closely related to chimpanzees. When combined with our preliminary data from adult male chimpanzees¹³, we hypothesized a comparatively greater degree of trabeculation in non-human great apes, which may be associated with a high degree of disorder in the fiber orientation of the trabeculated myocardium¹⁵. In turn, this may be associated with a change in the LV myofiber helix angle, which previous research has established is important in the generation of LV twist^{24,25}. Therefore, based on our preliminary data, we hypothesized some rotation, and therefore LV twist, would be present in chimpanzees (i.e. LV twist/apical rotation would never be 0.0%). Therefore, *a priori*, we did not hypothesize a constant ratio of change between left ventricular twist/apical rotation and the trabecular to compact ratio, and hence predicted a curvilinear relationship. Following data analysis, our prediction was confirmed, as a curvilinear relationship best fit our data, as opposed to linear.

References

1. Petersen, S. E. *et al.* Excessive Trabeculation of the Left Ventricle. *JACC: Cardiovascular Imaging* **16**, 408–425 (2023).
2. D’Silva, A. & Jensen, B. Left ventricular non-compaction cardiomyopathy: how many needles in the haystack? *Heart* **107**, 1344–1352 (2021).
3. Aung, N. *et al.* Prognostic Significance of Left Ventricular Noncompaction. *Circulation: Cardiovascular Imaging* **13**, e009712 (2020).
4. Anderson, R. H. *et al.* Key Questions Relating to Left Ventricular Noncompaction Cardiomyopathy: Is the Emperor Still Wearing Any Clothes? *Can J Cardiol* **33**, 747–757 (2017).
5. Chin, T. K., Perloff, J. K., Williams, R. G., Jue, K. & Mohrmann, R. Isolated noncompaction of left ventricular myocardium. A study of eight cases. *Circulation* **82**, 507–513 (1990).
6. Oechslin, E. & Jenni, R. Left ventricular non-compaction revisited: a distinct phenotype with genetic heterogeneity? *Eur Heart J* **32**, 1446–1456 (2011).
7. Faber, J. W., D’Silva, A., Christoffels, V. M. & Jensen, B. Lack of morphometric evidence for ventricular compaction in humans. *J Cardiol* **78**, 397–405 (2021).
8. Gati, S. *et al.* Reversible de novo left ventricular trabeculations in pregnant women: implications for the diagnosis of left ventricular noncompaction in low-risk populations. *Circulation* **130**, 475–483 (2014).
9. Gati, S. *et al.* Increased left ventricular trabeculation in individuals with sickle cell anaemia: physiology or pathology? *Int J Cardiol* **168**, 1658–1660 (2013).
10. Gati, S. *et al.* Increased left ventricular trabeculation in highly trained athletes: do we need more stringent criteria for the diagnosis of left ventricular non-compaction in athletes? *Heart* **99**, 401–408 (2013).
11. de la Chica, J. A. *et al.* Association Between Left Ventricular Noncompaction and Vigorous Physical Activity. *J Am Coll Cardiol* **76**, 1723–1733 (2020).

12. Woodbridge, S. P. *et al.* Physical activity and left ventricular trabeculation in the UK Biobank community-based cohort study. *Heart* **105**, 990–998 (2019).
13. Shave, R. E. *et al.* Selection of endurance capabilities and the trade-off between pressure and volume in the evolution of the human heart. *Proc Natl Acad Sci U S A* **116**, 19905–19910 (2019).
14. Faber, J. W. *et al.* Equal force generation potential of trabecular and compact wall ventricular cardiomyocytes. *iScience* **25**, 105393 (2022).
15. Ross, M. A. & Streeter, D. D. Nonuniform subendocardial fiber orientation in the normal macaque left ventricle. *Eur J Cardiol* **3**, 229–247 (1975).
16. Sallin, E. A. Fiber orientation and ejection fraction in the human left ventricle. *Biophys J* **9**, 954–964 (1969).
17. Kawel, N. *et al.* Trabeculated (Non-Compacted) and Compact Myocardium in Adults: The Multi-Ethnic Study of Atherosclerosis. *Circ Cardiovasc Imaging* **5**, 357–366 (2012).
18. Geerts, L., Bovendeerd, P., Nicolay, K. & Arts, T. Characterization of the normal cardiac myofiber field in goat measured with MR-diffusion tensor imaging. *Am J Physiol Heart Circ Physiol* **283**, H139-145 (2002).
19. Chen, J. *et al.* Remodeling of cardiac fiber structure after infarction in rats quantified with diffusion tensor MRI. *Am J Physiol Heart Circ Physiol* **285**, H946-954 (2003).
20. Healy, L. J., Jiang, Y. & Hsu, E. W. Quantitative comparison of myocardial fiber structure between mice, rabbit, and sheep using diffusion tensor cardiovascular magnetic resonance. *J Cardiovasc Magn Reson* **13**, 74 (2011).
21. Streeter Jr, D. D. Gross morphology and fiber geometry of the heart. *The cardiovascular system* 61–112 (1979).
22. Henson, R. E., Song, S. K., Pastorek, J. S., Ackerman, J. J. H. & Lorenz, C. H. Left ventricular torsion is equal in mice and humans. *American Journal of Physiology-Heart and Circulatory Physiology* **278**, H1117–H1123 (2000).

23. Arts, T., Meerbaum, S., Reneman, R. S. & Corday, E. Torsion of the left ventricle during the ejection phase in the intact dog. *Cardiovascular Research* **18**, 183–193 (1984).
24. van Dalen, B. M. *et al.* Influence of cardiac shape on left ventricular twist. *J Appl Physiol (1985)* **108**, 146–151 (2010).
25. Taber, L. A., Yang, M. & Podszus, W. W. Mechanics of ventricular torsion. *J Biomech* **29**, 745–752 (1996).
26. Atencia, R. *et al.* HEART RATE AND INDIRECT BLOOD PRESSURE RESPONSES TO FOUR DIFFERENT FIELD ANESTHETIC PROTOCOLS IN WILD-BORN CAPTIVE CHIMPANZEES (PAN TROGLODYTES). *J Zoo Wildl Med* **48**, 636–644 (2017).
27. Drane, A. L. *et al.* Cardiac structure and function characterized across age groups and between sexes in healthy wild-born captive chimpanzees (Pan troglodytes) living in sanctuaries. *Am J Vet Res* **80**, 547–557 (2019).

Reviewers' comments:

Reviewer #1 (Remarks to the Author):

Your rebuttal is exemplary, and the modifications to your initial manuscript respond appropriately to my own criticisms. We will need to wait to see whether my co-referee is also satisfied by your response.

I do, however, have two additional comments that I believe would make your work more meaningful for the ongoing investigations that you have rightly indicated will be needed to resolve the as yet unanswered questions you have highlighted.

In the first instance, throughout your text you refer to "myocardial fibers". This does not surprise me, since I have used the same terms myself. I am now of the opinion that the so-called "fiber" does not exist, at least not in my own understanding. In my own current opinion, with which you may well disagree, the working unit of the myocardium is the cardiomyocytes. The cardiomyocytes are then aggregated together to form a three-dimensional mesh, with components of this mesh producing the helical arrays to which you have rightly drawn attention. As yet, however, I know of no evidence to show that these components can be identified as individual entities warranting the description of "fibers". It would be better, surely, simply to account for the aggregated cardiomyocytes, or alternatively to offer a definition of your own understanding of the "myofiber".

The second problem lies with your depiction of the "bulls-eye" to show the ventricular components. I am aware that the stance of the apes does not parallel entirely that of humans. You label the figure, however, to show "anterior" as the antonym of "inferior". This cannot be correct, even if the terms are widely used by clinicians. The antonym of "inferior" is "superior". The rules of basic anatomy, therefore, point to the inaccuracy of the current labelling of the figure. You may be happy to accept "conventional wisdom", but this should not be permitted to undermine the rules of the English language.

Robert H. Anderson

Reviewer #2 (Remarks to the Author):

The revised manuscript is an improvement of the original submission.

Your replies;

1-2. No further comments.

3. It seems to me that the vast majority of the LV and RV myocardium is nourished by the same (coronary arterial) blood and that the transcriptional differences are small, i.e. the myocardium is much the same. Lumpers and splitters ... On L203-204 & 223-224; You write "proportionately greater compact myocardium may have enhanced myocardial rotation and deformation, and thus aided efficient ventricular filling and ejection" & "relaxation and contraction of the trabecular meshwork, might aid LV filling and ejection". These two quotes are of course not contradictory in the sense that any myocardium will impact on diastolic and systolic function. But this text leaves me rather confused as to what you think/propose should be the functional consequence of more or less compact vs trabeculated myocardium.

4. No further comments.

5. Nice!

6. I was not suggesting your data was inconsistent with previous reports. By borderline failing I meant the strain values you report would be suggestive of heart failure had they been found in human. If longitudinal strain, which correlates to MAPSE and EF (in human), is lower in chimpanzees than in human, and chimpanzee apical strain readouts are also lower than in human, how is this not indicative of borderline failing of the chimpanzee LV? Are we to believe that strain (in chimpanzees) is un-related to stroke volume? Drane et al 2019 report stroke volumes that are approximately 1/1000 ml of BM which is what one would expect of a mammal (meta-analysis by Seymour and Blaylock 2000). If strain is not

related to stroke volume, then I don't see the value of the comparison. It is also in this regard that I asked whether you have measurements of stroke volume (and heart rate = cardiac output) and ejection fraction. That would allow you to show how the 'anthropocentric low' chimpanzee strain values relate to what really matters to the animal; cardiac output.

7. Thank you for your explanation. I don't intend to be pedantic, but your answer hardly amounts to a rationale. (Suppose the choice of test was based on "best fit" as you write, you could achieve an even better fit (R^2) with a multi-factor polynomial). Mind you, I suspect you will find significant correlations if you did a linear regression analysis, judging from how your human data clusters differently from the other data.

Response to Reviewers

Your manuscript entitled "Left ventricular trabeculation in Hominidae: divergence of the human cardiac phenotype" has now been seen by 2 referees. You will see from their comments below that they recognize the improvements you made during revisions, but requested a few final clarifications. We are of course very interested in the possibility of publishing your study in Communications Biology, but would like to consider your response to these concerns in the form of a revised manuscript before we make a final decision on publication. We therefore invite you to revise and resubmit your manuscript, taking into account the points raised. In particular, please carefully address the minor clarifications and changes requested, making appropriate textual changes and adding any potential limitations of your experimental approach in the discussion section, if applicable.

We are grateful to the reviewers for their helpful suggestions and feedback throughout the review process. We have revised the manuscript, taking into consideration all comments, and genuinely believe their comments and suggestions have significantly improved the manuscript.

Reviewer #1

1. Your rebuttal is exemplary, and the modifications to your initial manuscript respond appropriately to my own criticisms. We will need to wait to see whether my co-referee is also satisfied by your response.

I do, however, have two additional comments that I believe would make your work more meaningful for the ongoing investigations that you have rightly indicated will be needed to resolve the as yet unanswered questions you have highlighted. In the first instance, throughout your text you refer to "myocardial fibers". This does not surprise me, since I have used the same terms myself. I am now of the opinion that the so-called "fiber" does not exist, at least not in my own understanding. In my own current opinion, with which you have well disagree, the working unit of the myocardium is the cardiomyocytes. The cardiomyocytes are then aggregated together to form a three-dimensional mesh, with components of this mesh producing the helical arrays to which you have rightly drawn attention. As yet, however, I know of no evidence to show that these components can be identified as individual entities warranting the description of "fibers". It would be better, surely, simply to account for the aggregated cardiomyocytes, or alternatively to offer a definition of your own understanding of the "myofiber".

We sincerely thank the reviewer for their very positive feedback, and appreciate their insightful suggestions and critique of the manuscript. We are grateful for all of their comments during the peer-review process, and believe the manuscript has significantly benefited from the reviewer's insight and knowledge in this area.

We appreciate the reviewer's helpful comment regarding the terminology, and agree with their suggestion that the term 'aggregated cardiomyocytes' as opposed to 'myofiber' would be more appropriate. We have therefore amended the text in both the introduction (lines 103-104) and discussion (lines 207-208 and 210).

2. The second problem lies with your depiction of the "bulls-eye" to show the ventricular components. I am aware that the stance of the apes does not parallel entirely that of humans. You label the figure, however, to show "anterior" as the antonym of "inferior". This cannot be correct, even if the terms are widely used by clinicians. The antonym of "inferior" is "superior". The rules of basic anatomy, therefore, point to the inaccuracy of the current labelling of the figure. You may be happy to accept "conventional wisdom", but this should not be permitted to undermine the rules of the English language.

The reviewer raises an important consideration, which we are grateful for. As the reviewer notes, we have adopted the standard clinical convention for the description of left ventricular segmentation. This follows the guidelines for cardiac chamber quantification provided by the American Society of Echocardiography and the European Association of Cardiovascular Imaging¹. However, we completely agree with the reviewer that the recognized convention in cardiology does not align with the basic rules of anatomical description. Unfortunately, whichever approach we adopt we may create some confusion. In order to acknowledge this, and to aid readers who may not be familiar with the recognized clinical nomenclature, we have included a footnote for Figure 1 (lines 569 – 573) to provide greater clarification:

‘Anatomical labels have been provided in accordance with the conventional guidelines for cardiac chamber quantification by the American Society of Echocardiography and European Association of Cardiovascular Imaging¹. However, we note that this clinical convention does not align with the recognized anatomical approach and may result in confusion across disciplines – see reference² for further clarification.’

Reviewer #2

The revised manuscript is an improvement of the original submission.

We thank the reviewer for their helpful comments and feedback throughout the peer-review process that has significantly contributed to improving the manuscript.

Your replies;

1-2. No further comments

3. It seems to me that the vast majority of the LV and RV myocardium is nourished by the same (coronary arterial) blood and that the transcriptional differences are small, i.e. the myocardium is much the same. Lumpers and splitters ... On L203-204 & 223-224; You write “proportionately greater compact myocardium

may have enhanced myocardial rotation and deformation, and thus aided efficient ventricular filling and ejection” & “relaxation and contraction of the trabecular meshwork, might aid LV filling and ejection”. These two quotes are of course not contradictory in the sense that any myocardium will impact on diastolic and systolic function. But this text leaves me rather confused as to what you think/propose should be the functional consequence of more or less compact vs trabeculated myocardium.

We thank the reviewer for their comment; however, unfortunately we do not fully understand the reviewer’s concern. In the latest revision, we addressed both reviewers concerns and specifically developed the connection between the helical angulation of the aggregated cardiomyocytes and how this may support the ventricular mechanics that underpin function. Our suggestion with regard to a potential functional consequence of a more compact myocardium, as observed in humans, is that this may aid ventricular filling/ejection through enhanced left ventricular mechanics. This is emphasized in lines 210-214 where we conclude *“If this non-uniform structural arrangement is also present in the trabeculated LV in non-human great apes, this may help to explain the lower rotation and deformation observed in these species (Figure 3). However, further research is needed to specifically determine the myocardial architecture of the LV in non-human great apes, and its relationship with myocardial function”*.

With regard to the potential functional consequences of a more trabeculated myocardium, as observed in the non-human great apes, we have been deliberately cautious not to overinterpret our data. However, the “opening” and “closing” of intertrabecular recesses during diastole and systole, respectively, may create interventricular pressure gradients (noted in lines 222-224) that aid filling and ejection. We believe that suggesting anything more than this, from the current dataset, would be inappropriate. Notwithstanding, the evident species-level differences in the degree of compact versus trabeculated myocardium and ventricular mechanics provide a basis for future work to investigate the functional consequence of these parameters, which we acknowledge in our discussion (lines 228-230). Hopefully, the reviewer will agree that, based on the data presented in the manuscript, we have provided some logical explanations as to the potential functional consequence of a more, or less, compact versus trabeculated myocardium, and that to speculate further would be counterproductive without additional data.

4. No further comments.

5. Nice!

6. I was not suggesting your data was inconsistent with previous reports. By borderline failing I meant the strain values you report would be suggestive of heart failure had they been found in human. If longitudinal strain, which correlates to MAPSE and EF (in human), is lower in chimpanzees than in human, and chimpanzee apical strain readouts are also lower than in human, how is this not indicative borderline failing

of the chimpanzee LV? Are we to believe that strain (in chimpanzees) is un-related to stroke volume? Drane et al 2019 report stroke volumes that are approximately 1/1000 ml of BM which is what one would expect of a mammal (meta-analysis by Seymour and Blaylock 2000). If strain is not related to stroke volume, then I don't see the value of the comparison. It is also in this regard that I asked whether you have measurements of stroke volume (and heart rate = cardiac output) and ejection fraction. That would allow you to show how the 'anthropocentric low' chimpanzee strain values relate to what really matters to the animal; cardiac output.

We thank the reviewer for their comment and appreciate the opportunity to continue this nuanced discussion. Importantly, while the stroke volume data reported in our previous manuscript (Drane et al., 2019³) generally fall in line with other mammalian data (e.g. Seymour and Black, 2000), Shave et al., 2019 demonstrated that cardiac output in adult male chimpanzees is approximately 25% lower than that of adult male humans⁴. As such, the overall absolute "requirement" of the left ventricle to generate stroke volume is possibly lower in chimpanzees in comparison to humans. Therefore, it is possible that the lower left ventricular mechanics (i.e. myocardial deformation and rotation) observed in the present manuscript (likely alongside other non-examined contributory factors/mechanisms), are adequate to generate the required stroke volume in non-human great apes (i.e. lower absolute mechanics do not represent a failing ventricle in these species).

As noted in our previous response, we are reluctant to re-publish data. However, to address the reviewer's concern regarding the potential relationship between our left ventricular mechanics data and global functional parameters, we have now included the relationships between ejection fraction, longitudinal strain and left ventricular twist as supplementary material. These data confirm that the relationships that have been previously described in humans⁵ also exist in chimpanzees (albeit with lower absolute left ventricular mechanics values). We have amended the text on lines 349 - 352, and incorporated a figure in the Supplementary Information to reflect this:

'To confirm that LV rotation and deformation are related to global measures of cardiac function in chimpanzees, as has been reported in humans, we explored the relationship between ejection fraction, and LV longitudinal strain and LV twist (Supplementary Fig. 3).

We trust that these data help the reviewer reconcile the value in comparing the left ventricular mechanics parameters across species. We also acknowledge that with a larger human sample we may be able to specifically explore, across species, the relative contribution of each of the mechanical parameters to the overall stroke volume (and the constituent end-diastolic and end-systolic volumes), and indeed that is something we intend to do in the future. However, we believe that this is beyond the scope of the current study, which specifically aimed to compare trabeculations and ventricular mechanics across species.

7. Thank you for your explanation. I don't intend to be pedantic, but your answer hardly amounts to a rationale. (Suppose the choice of test was based on "best fit" as you write, you could achieve an even better fit (R2) with a multi-factor polynomial). Mind you, I suspect you will find significant correlations if you did a linear regression analysis, judging from how your human data clusters differently from the other data.

We thank the reviewer for their comment. The relationships between the apical T:C ratio and the LV rotation and deformation parameters were part of an exploratory analysis to begin to unpick the functional consequence of a more compact or trabeculated myocardium. Based on our rationale (as outlined in our previous response), we did not predict a constant ratio of change between our parameters of interest. We predicted that there would be a point at which having more compact myocardium would not be associated with any greater deformation/rotation; as there would also be a point at which a more trabeculated myocardium would not be associated with any further reduction in deformation/rotation, hence exponential plateau curves were chosen. Based on the prior suggestions from the reviewer, we did examine the data using a linear regression, but in line with our *a priori* prediction, the exponential plateau curves had the best fit to the data (based on sum-of-squares). We did not *a priori* consider a multi-factorial polynomial, and we do not think it appropriate to simply apply different models to find the best fit. Hopefully, our hypothesis driven, *a priori* approach is deemed appropriate.

References

1. Lang, R. M. *et al.* Recommendations for cardiac chamber quantification by echocardiography in adults: an update from the American Society of Echocardiography and the European Association of Cardiovascular Imaging. *J Am Soc Echocardiogr* **28**, 1-39.e14 (2015).
2. Mori, S., Tretter, J. T., Spicer, D. E., Bolender, D. L. & Anderson, R. H. What is the real cardiac anatomy? *Clin Anat* **32**, 288–309 (2019).
3. Drane, A. L. *et al.* Cardiac structure and function characterized across age groups and between sexes in healthy wild-born captive chimpanzees (*Pan troglodytes*) living in sanctuaries. *Am J Vet Res* **80**, 547–557 (2019).
4. Shave, R. E. *et al.* Selection of endurance capabilities and the trade-off between pressure and volume in the evolution of the human heart. *Proc Natl Acad Sci U S A* **116**, 19905–19910 (2019).
5. Onishi, T. *et al.* Global Longitudinal Strain and Global Circumferential Strain by Speckle-Tracking Echocardiography and Feature-Tracking Cardiac Magnetic Resonance Imaging: Comparison with Left Ventricular Ejection Fraction. *Journal of the American Society of Echocardiography* **28**, 587–596 (2015).